# Lipsticks History, Formulations, and Production: A Narrative Review

**Saeid Mezail Mawazi** [1,*], **Nurul Aqilah Binti Azreen Redzal** [1], **Noordin Othman** [2,3]
and **Sultan Othman Alolayan** [2]

1    Department of Pharmaceutics, School of Pharmacy, Management and Science University, University Drive,
     Off Persiaran Olahraga, Shah Alam 40100, Selangor, Malaysia; 012018090595@sp.msu.edu.my
2    Department of Clinical and Hospital Pharmacy, College of Pharmacy, Taibah University,
     Al-Madinah Al-Munawwarah 30001, Saudi Arabia; nbinothman@taibahu.edu.sa (N.O.);
     solayan@taibahu.edu.sa (S.O.A.)
3    Department of Clinical Pharmacy, School of Pharmacy, Management and Science University, University Drive,
     Off Persiaran Olahraga, Shah Alam 40100, Selangor, Malaysia
*    Correspondence: saeid_mezail@msu.edu.my

**Abstract:** A considerable amount of literature has been published on several aspects of lipsticks production. To date, there is no collation of studies related to lipsticks production that has been published. This review was conducted to examine information about the history of lipsticks; ingredients used in the preparation of lipsticks, focusing on the natural and chemical ingredients; methods of preparation for the lipsticks; and the characterization of the lipsticks. A literature search for English language articles was conducted by searching electronic databases including Web of Science, Scopus, PubMed, and Google Scholar. Overall, the evidence indicates that lipsticks have been used since ancient times and are among the highest demand cosmetics. The findings of this review summarize those of earlier studies that explained the use of different types of ingredients in the manufacturing processes of lipsticks. It highlights the importance of using green technology and ingredients to fabricate lipsticks to avoid potential side effects such as skin irritation and allergy reaction.

**Keywords:** lipsticks; lipstick's formulations; cosmetics; characterization of lipsticks; lipsticks methods of preparation

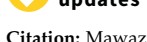



## 1. Introduction

Lipsticks are one of the most widely used cosmetic products. Social, psychological, and therapeutic benefits can be attained from using lipstick [1]. The beauty and attractiveness of a person are enhanced as lipsticks colour the lips and protect them from the external environment. However, current lip care products not only emphasize aesthetic value but also preferably have added medicinal value to the lip of consumers. This led to the emergence in the market of medicated lipsticks with active medicinal ingredients. The medicated lipsticks may provide protection against infections of bacteria due to the presence of an active medicinal ingredient in the formulation. This function adds on to the existing role of lipsticks, which provide moisture and emollient action to prevent cracking and chapping of the lips [2].

Recently, the use of herbs in the production of cosmeceuticals products for personal care has been on the rise [3]. Herbal cosmetics, also known as natural cosmetics, are the modern trend which encircles both health and beauty care [4]. These products are experiencing more demand in the market, as nowadays most people prefer natural over chemical products [5]. Natural cosmetics not only provide satisfaction, as these products are free from synthetic chemicals and have relatively fewer side effects, but also supply the body with nutrients and enhance health of a person [6].

On some occasions, regular usage of synthetic products that contain ingredient such as lead, petrolatum, and phthalates may be harmful to consumers. These products may

cause lip irritations and dry, chapped lips and can lead to health problem such as allergies, asthma, and cancer [7,8]. The habits of licking lips or eating and drinking while wearing lipstick can worsen the problems. Due to chronic exposure and their neurotoxic nature, harmful chemicals such as lead in lipsticks should not be ignored [1,9]. Therefore, the aim of the present review is to look at the data on lipsticks, with the goal of providing information on the history of lipsticks, materials used in lipstick preparation (with an emphasis on natural and artificial ingredients), preparation methods, and characterization.

## 2. History of Lipstick

Men began using colours for adornment in approximately 3000 BC in order to attract the animals they wanted to hunt. Generally, the concept and construct of "cosmeceuticals" was first articulated by Raymond Reed (1961), the founder of the US Society of Cosmetic Chemists [10]. It originated from the Greek term "kosm tikos", which means "decorating talent" [10]. Later, in 1984, Albert Kligman used the word "cosmeceuticals" referring to the compounds that have both cosmetic and medicinal properties [10]. Many herbs and floras have been used in the manufacture of cosmetics for the purposes of beauty and protection from external influences. The natural chemicals in cosmetics do not harm the human body; instead, they provide it with nutrients and minerals [10]. Lipsticks, in particular, have been used by humans for over 500 years [11]. Lipstick was first discovered as a rough fragment of brick in ancient Mesopotamia [11]. Colouring lips is an ancient tradition that dates to the prehistoric period [11]. Lipstick was first introduced in France in 1869 as a cosmetic product made from animal fat and beeswax [11]. The availability of lipstick in the form of cylindrical metal tubes was introduced in 1915 [12]. Presently lipsticks have become an essential product for many consumers. There is an extensive choice of colour shades and textures. This can be observed from the fact that lipstick is being marketed in hundreds of shades of colours to satisfy the increasing demand [13].

## 3. Ingredients of Lipstick

Lipsticks contain wide range of the ingredients made from natural sources, chemical sources, and a mix of both. Nevertheless, synthetic-based lipstick ingredients and natural-based lipstick ingredients are available in the market. Using synthetic-based ingredients of lipsticks may produce a serious adverse reaction [9]. For example, the presence of lead in lipsticks and colouring ingredients is one of the most serious issues [9]. Nickel and copper, two metals commonly found in cosmetics, can trigger allergic reactions in certain people [9]. Common ingredients used to formulate herbal lipsticks include (Tables 1 and 2) castor oil, paraffin wax, beeswax, beet root juice, ripe fruit powder of *shikakai*, lemon oil, orange essence, and vanilla essence [14]. However, previous studies indicate that there is a minor difference regarding the ingredients used in the production of lipsticks.

**Table 1.** Synthetic ingredients with their quantities in the formulation of lipsticks.

| No. | Ingredients | Functions | Quantity % (*w/w*) | References |
|---|---|---|---|---|
| 1 | Paraffin wax | Glossy, hardness, stiffening agent | 28 | [11,14–17] |
| 2 | Butyl stearate | Lipstick base and solvent for dyestuff and dispersing agent | 1–25 | [18–20] |
| 3 | Microcrystalline wax | Lipstick base | 2 | [21] |
| 4 | Ozokerite wax | Lipstick base | 3–10 | [20,21] |
| 5 | Ceresin wax | Lipstick base or to increase the melting point of other waxes | 3–10 | [20,22,23] |
| 6 | Oleyl alcohol | Blending agent, emollient, oleaginous vehicle, solvent. | 40–50 | [20,24] |

**Table 1.** *Cont.*

| No. | Ingredients | Functions | Quantity % (*w/w*) | References |
|-----|-------------|-----------|---------------------|------------|
| 7 | Methyl paraben | Preservative | 0.1–1 | [25] |
| 8 | Propyl paraben | Preservative | 0.1–1 | [25] |
| 9 | Propyl-p-hydroxybenzoate | Preservatives | 0.1–0.2 | [20,26] |
| 10 | Vitamin E | Antioxidant | 0.5 | [27,28] |
| 11 | Lanolin alcohol | Blending agents and plasticizing effect | 2–5 | [20,21,25,29] |
| 12 | Anhydrous lanolin | Blending agent | 2–20 | [20,30] |
| 13 | Titanium dioxide | Pigment shading agent, brightener | 1–40 | [20,31,32] |
| 14 | Zinc oxide | Pigment, brightener | 1–40 | [20,31,32] |
| 15 | Calcium, barium, and aluminium lakes | Colouring agents | 10–15 | [20,33] |
| 16 | Isopropyl myristate or isopropyl palmitate | Glossing agent | 2–3 | [20,34] |
| 17 | Acetoglycerides | Blending agents and plasticisers | 2.5–7 | [20,35,36] |
| 18 | Bromo mixture | Colouring agents | 2–25 | [24,37] |

**Table 2.** Natural ingredients with their quantities in the formulation of lipsticks.

| No. | Ingredients | Functions | Quantity (%) | References |
|-----|-------------|-----------|--------------|------------|
| 1 | Ripe fruit powder of *Shikakai* | Surfactant | 12 | [14,15,38] |
| 2 | Lemon oil | Antioxidant, preservative, flavouring agent | 0.1–1 | [3,6,14–16,38–40] |
| 3 | Orange essence | Flavouring agent | 1.5 | [14,15] |
| 4 | Mango butter from *Mangifera indica* | Lipstick base | 10 | [41] |
| 5 | Beetroot juice | Colouring agent | 06 | [14–16] |
| 6 | *Theobroma cocoa* | Colouring agent | 40 | [27] |
| 7 | Lycopene from *Solanum lycopersicum L* (Tomato) | Colouring agent | 2.5 | [6,42] |
| 8 | *Punica granatum* from pomegranate | Colouring agent | 5–9 | [43–46] |
| 9 | *Amaranthus Cruentus* L. | Colouring agent | 0.5–1 | [39] |
| 10 | Jati leaves (*Tectona grandis* L.f.) | Colouring agent | 18–22 | [47] |
| 11 | Ginger powder | Antimicrobial agent | 2 | [45] |
| 12 | Turmeric Powder | Antimicrobial agent | 5–6 | [48] |
| 13 | *Hylocereus polyrhizus* | Antimicrobial agent and colouring agent | 4 | [49] |
| 14 | Vanilla essence | Preservative | 10 | [14,38,39,48,50] |
| 15 | Olive oil | Blending agent and lipstick base | 10–30 | [11,16] |

**Table 2.** *Cont.*

| No. | Ingredients | Functions | Quantity (%) | References |
|---|---|---|---|---|
| 16 | Castor oil | Blending agent, emollient, oleaginous vehicle, solvent. | 40–50 | [3,6,14,15,17,20,51,52] |
| 17 | Meadowfoam seed oil | Blending agent | 5 | [21] |
| 18 | Beeswax | Glossy, hardness, emollient. | 3–10 | [1,3,6,14–16,38,39,50–54] |
| 19 | Candelilla wax | Lipstick base and moisturizer | 1–10 | [55,56] |
| 20 | Carnauba wax | Lipstick base and moisturizer | 1–5 | [55,56] |
| 21 | Alkenones wax | Lipstick base and moisturizer | 2–5 | [21] |
| 22 | Coconut oil | Lipstick base and moisturizer | 25–45 | [50,55,57] |
| 23 | Pitaya (*Hylocereus polyrhizus*) seed oil | Lipstick base and moisturizer | 10–25 | [55] |
| 24 | Mangosteen rind (*Garcinia mangostana* L.) | Antioxidant | 4–8 | [25] |

Synthetic and natural waxes are a dominant feature of lipstick fabrication. To generate a sufficient film when the stick is applied to the lips, the oil combination must blend closely with the waxes [20]. The wax mixture's composition is quite important. Using a mixture of waxes with varying melting points and controlling the ultimate melting point of the stick by adding an appropriate amount of a high melting point wax yields the best results. Paraffin wax is one of the phase change materials. It has a high heat storage capacity, is readily available, and is inexpensive [58]. Slack wax is obtained in the initial step of paraffin wax manufacturing by solvent dewaxing vacuum distillates followed by chilling and filtration. This procedure makes use of an industrial vacuum-rotary filter. After filtration, the slack wax is separated from the remaining solvent, fractioned, and refined into white paraffin wax. As a result, the capacity of fossil paraffin wax production is inextricably related to lubricant production and, more broadly, crude oil refining [59]. A four-week study was conducted to investigate the effect of the paraffin wax mask pack on the skin of 20 healthy males and females. As a result, it was concluded that the paraffin wax mask pack had favourable effects on skin improvement and would be of great utility and value in the development of skin care devices [60]. In lipsticks, paraffin wax was used as a glossening, stiffening, and hardening agent [11,14–17]. To produce a glossy appearance after application, liquid paraffin or white mineral oils are also employed [20].

There are two forms of paraffin waxes: macrocrystalline wax and microcrystalline wax, both of which have significant physical differences in various aspects. Microcrystalline wax is opaque, plastic, malleable, and sticky, whereas macrocrystalline wax is translucent, glossy, slippery, and brittle [61]. Microcrystalline wax is derived from heavy distillates and used in a variety of applications such as cosmetics, rubber compounds, candles, and metal casting to take advantage of its unique flexibility, viscosity, temperature tolerance, and adhesive properties [61]. Recent evidence suggests that the microcrystalline wax was utilized in lipsticks production as a lipstick base and stiffening agent [62,63]. Ozokerite wax is manufactured from coal and shale [64]. It helps to raise the melting point of the stick [20]. Lipsticks containing more than 10% ozokerite have a tendency to crumble when applied [20]. Ceresin wax is also known as mineral wax, and it refers to a type of ozokerite that has been refined using sulphuric acid. Ceresin wax is now a generic term for commercial goods that combine pure ozokerite with other solid hydrocarbons to produce

waxes with varying melting points. It is comparable with ozokerite in that it is used to raise the melting point of a product [20,22,23].

Butyl stearate is a useful substance to utilise with castor oil. It works as a partial solvent and possesses wetting qualities, allowing undissolved dyestuff to disperse finely [20,37].

In some lipstick formulations, oleyl alcohol is utilised to substitute castor oil. The substance is a superior eosin solvent to castor oil and can be used to make high-staining sticks. If the waxes are adjusted to provide the product a high melting point, a stick based on oleyl alcohol distributes easily and evenly but leaves an oily film. To prevent rancidity, an antioxidant should be included in the lipstick formulations containing oleyl alcohol [24]. However, lipstick dermatitis caused by oleyl alcohol was originally documented in 1960, and there have been sporadic reports of allergic responses to oleyl alcohol since then, mostly in cosmetics. Because it is not routinely tested, oleyl alcohol may be a more frequent allergy than the few reports suggest [65]. Tosti et al. (1996) evaluated 146 individuals with a potential allergy to topical treatments containing five fatty alcohols and identified 34 positive responses, 33 of which were caused by oleyl alcohol (30%) in petrolatum. All of the controls were negative when we tested with 10% petrolatum [66].

A number of studies have begun to highlight the safety of the natural preservatives compared with synthetics [67–70]. It was suggested that a high antimicrobial efficacy could prolong the shelf-life of products [70]. However, synthetic preservatives have a high risk of causing adverse reactions to skin [70]. Parabens (PHB) are the most widely used synthetic preservatives in cosmetic products [70]. Contact dermatitis has been documented as a hypersensitivity reaction to parabens [71]. They have weak oestrogen-like properties, suggesting a correlation between breast cancer and parabens, and often cause skin irritation and provoke allergy reactions [70]. Among the cosmetics, lipsticks were reported to have the highest concentrations of parabens [72]. For example, antimicrobial preservatives such as methylparaben and other parabens are frequently employed in cosmetics and in oral and topical pharmaceutical formulations. Moreover, lipstick may also contain 0.1 percent propyl-p-hydroxybenzoate as preservative. Concentrations of 0.2 percent of propyl-p-hydroxybenzoate can elicit a minor burning sensation and, on rare occasions, an eosin allergic reaction on sensitive skin [20].

Evans and Bishop discovered vitamin E as an essential dietary element for rat reproduction in 1922 [73]. Vitamin E is available in a natural form and synthetic form (*α-tocopherol*) [74]. However, there has been no detailed investigation of using natural vitamin E or the synthetic one in the fabrication of lipsticks. Therefore, vitamin E has been used in lipstick preparations as an antioxidant [27,28].

Lanolin is utilised in cosmetics and topical medications because of its emollient qualities. Blending with anhydrous lanolin (wool fat) is typical, and the quantity utilised can range from 2% to 20%. A high lanolin content is beneficial when a specific emollient effect is desired or when a thick, unctuous film is preferred. Sticks with a high lanolin content might be oily or sticky, and the fragrance can be prominent, especially during storage. A lipstick may be made by combining the ingredients in the right amounts. It is more common to use fatty materials as blending agents, such as lanolin or one of the lanolin derivatives. Plasticizing is a term used to describe the action of these materials. They increase the thickness and durability of the film while improving its spreading characteristics. They also prevent liquid materials from separating from solids, which might result in sweating or blooming in the final product [20,21,25,29,30]. Nonetheless, despite the benefits, after repeated or prolonged use, lanolin may cause contact allergy in 1.2% to 6.9% of dermatitis patients [75–79].

Titanium dioxide is used in lipsticks as pigment or to alter the colour of the basic pigments. It has a high degree of brightness, which could give it a covering power over other white pigments [21,31,32]. When it comes to covering power, it outperforms and is preferred over zinc oxide in lipsticks and other cosmetic items [20]. It was originally used to create vibrant effects with high colour proportions, but it is now widely employed

with low colour proportions to create delicate pastel tones, while maintaining the required degree of opacity [20].

The primary colouring elements, as opposed to the staining materials, are insoluble dyestuffs and lake colours such as calcium, barium, and aluminium lakes. Depending on the tint and opacity of the film, the amount utilised in a lipstick varies between 10% and 15% [18,20]. To guarantee a smooth application, colours must have a fine and consistent particle size. They must have appropriate colouring and covering strength, as well as good opacity. The features of the colours' behaviour with oil are also important because they influence the ultimate uniformity of the mass [20]. Two or three percent isopropyl myristate or isopropyl palmitate produces a similarly effective gloss, and this concentration has no effect on the film's durability [20]. However, one of the primary concerns is the presence of lead in lipsticks and colouring chemicals [9].

Acetoglycerides are blending agents that alter the rheological properties of oils, fats, and waxes in lipstick formulations. They add plasticity properties to the stick formulations, allowing them to remain solid in hot weather and to maintain spreadability qualities at low temperatures [20,35]. For the purposes of formulation, they can be divided into two categories: liquid and solid. If the liquid acetoglycerides have a plasticizing effect on the oily ingredients, and the solid acetoglycerides have a comparable impact on the waxes, their behaviour may be explained. As a consequence, a mixture of two parts solid acetoglyceride and one part liquid acetoglyceride will provide a better outcome than utilising them separately [18,20,36].

The phrase "bromo mixture" refers to the component of the product that leaves an indelible stain, as opposed to the opaque film of colour produced by insoluble pigments. Bromo mixture is a solution that consists of dyestuff (also known as bromo acids) for staining in combination with appropriate ingredients [24]. The dyestuffs examples are fluoresceins, halogenated fluoresceins, and related water-insoluble dyes [24]. Bromo acids available in D and C colours are divided into two groups: red bromo acids that produce a red or reddish-blue stain and orange-red bromo acids that produce a pink to yellowish-pink stain [80]. Even though the undissolved dyestuff is finely disseminated, it stays suspended in the oil/wax film if castor oil is the only solvent present in the formula. The undissolved dyestuffs have a mild abrasive impact and the potential for allergic consequences if they are not detected and present in the finished product [24,37]. Bromo acid solvents such as tetrahydrofuryl alcohol and esters such as acetate, stearate, and benzoate are effective, albeit some of the esters, particularly the acetate, have penetrating odours that must be covered with appropriate scents [20]. Getting a suitable solvency lipstick base for bromo has always been a challenge [24]. Therefore, solvents of this sort tend to dry out the skin and can cause dermatitis, but because they dissolve up to 25% of bromo acids, only small amounts are needed to achieve good staining results. In this case, the quantity of emollient should be increased to counteract any drying effects [24].

Ripe fruit powder of *Shikakai* is a medicinal plant from which the fruits of this plant are used as surfactant in the formulation of lipsticks [38,81]. The finding of the McClements et al. (2017) is consistent with Schreiner et al. (2020), which noted that the use of natural surfactants encourage the production of environmentally friendly formulations, in line with the lower toxicity of such compounds [82,83].

Lemon oil is a colourless or yellow liquid and has a strong scent of lemon. The oil has an aroma of citrus [40]. It is used for flavouring lemon and exhibits some benefits such as antioxidant, anti-aging, and antimicrobial properties that are effective for bacterial and fungal infections [40]. Thus, the incorporation of lemon oil in lipsticks is essential for enhancing the formulation quality [6,16,38,40].

Flavouring agents play an important role in the taste-masking of a lipstick's ingredients. Kadu et al. (2015) and Sharma (2018) highlighted that the flavouring agents are an essential component to mask the odour of the fatty or wax base as well as imparting an attractive flavour [84,85]. Sainath et al. (2016), Bhagwat et al. (2017), and Rasheed et al. (2020) used strawberry essence [6,13,86], and Sunil et al. (2013) used orange essence [14].

Mango butter, also known as Amra, manga, or mango, is a tropical fruit derived from *Mangifera indica* and belongs to the Anacardiaceae family. It contains a wide spectrum of therapeutic benefits in practically every part of the plant. The extracts of *M. indica* have been reported to have antiviral, antibacterial, analgesic, anti-inflammatory, immunomodulatory, anti-amoebic, cardiotonic, and diuretic effects [41].

Both natural and synthetic colourants are used in lipstick. Synthetic colourants may cause adverse health effects. Natural colourants are safer and provide additional benefits, including antioxidant activity [87]. Sunil et al. (2013) and Chaudhari et al. (2019) utilized beet root juice as a colourant in the formulation of lipsticks [14,16]. Beetroot (*Beta vulgaris*) is the main source of natural red dye, called "beetroot red" [86]. Betanine is the main part of the red colorant extracted from common beetroot [86]. The roots are most typically deep red to purple in colour due to a variety of betalain pigments [86]. Different quantities of beetroot result in different colours of lipstick. Chaudhari et al. (2019) formulated three lipsticks with different quantities of beetroot; 3 g: pink, 7 g: dark red, 5 g pinkish red [16]. Sunil et al. (2013) only formulated a lipstick using 6 g of beetroot, resulting in a red colour lipstick [14]. Additionally, Chaudhari et al. (2019) used decoction process involving beetroot with ethanol for the extraction of colour pigment, while Sunil et al. (2013) did not report the extraction process [14,16]. Several studies used turmeric powder and 2% turmeric extract, which results in a yellow colour lipstick [27,38]. Different degrees of the brown colour in the lipstick could be obtained by the inclusion of cocoa bean (*Theobroma cocoa*) during the formulation processes [27]. The tree *Theobroma cacao* L. is a small in size tree and 4–8 m tall evergreen of the *Sterculiaceae* family that is endemic to the tropical Americas. Cocoa beans are high in carbohydrates (31%), protein (11%), fat (54%), fibre (16%), and minerals [88]. They consist of polyphenols, which include both flavonoids and non-flavonoids, and are the most bioactive components possessing antioxidant and anti-inflammatory properties [88].

Lycopene is a naturally occurring carotenoid that gives tomatoes, rosehips, watermelon, and pink grapefruit their red colour [89]. Carotenoids have been linked to a lower risk of degenerative illnesses in epidemiological research. On the one hand, lycopene has been shown to have a variety of pharmacological and nutritional effects in animals and humans, as well as promising antioxidant advantages [89]. There is a relatively small body of literature that presents the use of lycopene as a herbal lipstick ingredient [13,90].

Due to its potential biological applications, such as antioxidant, anticarcinogenic, and anti-inflammatory properties, pomegranate constituents (*Punica granatum*) could be employed as a desirable ingredient for the formulations lipsticks [91]. A considerable amount of literature has been published on *Punica granatum* herbal-based lipstick. These studies have revealed the benefits of using *Punica granatum* as a natural colouring agent in the herbal-based lipsticks [43–46]. Because it contains anthocyanin pigment, jati leaves (*Tectona grandis* L.f.) are plants of the *Verbenaceae* family that can be used as a natural dye [47]. Anthocyanins have different colours based on the type of plant. They are available as blue, purple, violet, magenta, red, and orange colours [47]. Jati leaves (*Tectona grandis* L.f.) are suggested to be used a pigment in the preparation of natural lipsticks in concentrations ranging from 18% to 22% [47]. Anthocyanins could be extracted also from grapes, blueberry, plum, purple cabbage, and blackberry [92].

Natural preservatives have been documented to be used in the preparation of lipstick, such as tea tree, lemon grass, rosemary, lavender, and ginger powder [45,70]. As an alternative to parabens, ginger (*Zingiber officinale*) has been widely tested for its antimicrobial activity [93–95]. It is a common plant in the Zingiberaceae family that is widely grown in China's central, southeastern, and southwestern regions, as well as throughout tropical Asia [96]. Much of the current literature on ginger pays particular attention to isolating and analysing pharmacologically a variety of bioactive compounds such as tannins, flavonoids, glycosides, essential oils, furostanol, spirostanol, saponins, phytosterols, amides, and alkaloids from different parts of the plants [95]. One longitudinal study found that ginger extract showed a potent antimicrobial activity against food-borne pathogens such as *Pseudomonas aruginosa*, *Staphylococcus aureus*, *Klebsiella* spp., *Vibrio cholerae*, *Escherichia coli*, and

*Salmonella* spp. [93]. Although some research has been carried out on the investigation of ginger antimicrobial activity, no studies have been found specifically for the investigation of antimicrobial activity of ginger in a lipstick formulation.

*Curcuma longa* (turmeric) was utilised as a spice in food and as a medicinal plant for a variety of ailments, including inflammation, pain, wound healing, and digestive issues. Turmeric and its bioactive curcuminoid polyphenols have been shown to impact a range of chronic diseases in preclinical studies [97]. A spectrum of antimicrobial activity was concluded for different fractions of turmeric against *Staphylococcus aureus*, suggesting the use of turmeric for the management of microbial infections [98]. However, several studies have used turmeric powder in the formulation of lipstick as an antimicrobial agent [99]. It has also been suggested as a natural colouring agent in the manufacturing of lipstick [27]. The Cactaceae family's *Hylocereus polyrhizus*, sometimes known as dragon fruit, has been used as a red colouring agent in lipstick preparations [49]. It was found to have potential antimicrobial activity in natural lipstick formulations when tested against the growth of *Escherichia coli*, *Klebsiella pneumonia*, *Salmonella typhimurium*, *Staphylococcus aureus*, and *Enterococcus faecalis* [2].

Vanilla essence has been utilized as a preservative in lipstick formulation [50]. Vanilla essence could be extracted from the vanilla pod of *Vanilla planifolia*. *Vanilla planifolia* belongs to the Orchidaceae family and is a perennial hanging plant native to tropical rainforests in Mexico, as well as in Madagascar, Tahiti, Indonesia, Seychelles, and the Philippines [100]. Vanilla has a long list of health advantages, such as antioxidant, antineoplastic, and cholesterol lowering effects; anti-sickling activity; and antimicrobial activity against *Staphylococcus aureus*, *Staphylococcus epidermidis*, *Bacillus cereus*, *Escherichia coli*, and *Yersinia enterocolitica* [101]. It has the ability to suppress peroxynitrite-mediated processes, which are significant in neurodegenerative illnesses including Alzheimer's and Parkinson's [101]. The vast majority of studies on using vanilla essence in lipstick formulations have been quantitative. Based on the reviewed data, there is no single study that has investigated the preservation effects of vanilla essence in lipstick.

Olive oil is one of the major components of olive fruit, which are grown on the olive tree (*Olea eumpaea*). Oleic acid, phenolic compounds, and squalene are the main active ingredients in olive oil. The phenolics include hydroxy tyrosol, tyrosol, and oleuropein, which are found in the virgin olive oil and have been shown to have antioxidant properties [102]. Phenolics can be used to reduce the incidence of coronary heart disease and hypertension; reduce the risk of some cancers, including as breast, skin, and colon cancer; and to present antimicrobial activity and anti-inflammation activity [102]. Olive oil has been used as a blending agent in the formulation of lipsticks [16,39]. It has also been utilised as a base for the preparation of water in olive oil (W/O) types of lipsticks [11]. Sainath et al. (2016), Patil et al. (2019), and Karanje et al. (2020) used olive oil in their study [29,39,86]. However, there was no further discussion about the role of olive oil in their lipstick formulation. A study by Viola and Violab (2009) mentioned that the phenol structure of olive oil has shown antioxidant action, especially oleuropeine, which acts against free radicals at the skin level [103]. Olive oil has an inhibitory effect on sun-induced cancer development when applied on the skin after the sun exposure [103]. On the other hand, Gorini et al. (2019) indicated that topical treatment with olive oil has a detrimental effect on skin barrier function and has the potential to promote the development of and to exacerbate existing atopic dermatitis [3,104].

Castor oil, also known as ricinus oil, is obtained from the seeds of the castor oil plant *Ricinuscommunis* and characterized as colourless to very pale yellow [15]. The high viscosity of castor oil could prevent lipsticks from smearing off, enhance the stability towards oxidation degradation, and be compatible with other ingredients [3,6,15,51]. Castor oil was commonly utilised in hairstyling. Today, it is widely used as a lipstick base because of its viscosity and texture. In addition, because of its high solvent power for bromo acids, it is occasionally employed as lipstick stains [37].

Beeswax is a natural wax from honeybees of the genus *Apis,* and it is recognised as a mandatory ingredient in the formulation of lipstick [15]. It is used as a glazing agent for a glossy look as well as to harden the texture of the lipstick [15,16,38,39,51,105]. Beeswax also helps to retain moisture for dry and chapped lips [54]. Various research studies have also discovered that beeswax contains small amounts of natural antibacterial agents and can help prevent painful inflammation that comes with an infection [106,107]. However, there was a slight difference in the quantity of beeswax and castor oil from the study by Aher et al. (2012), where the quantity for beeswax used was 15 grams, castor oil was 36 grams, while Sunil et al. (2013) used 36 grams of beeswax and 16 grams of castor oil [14,105]. Both results reported a stable formulation of lipstick [14,105]. However, no research article focused on the quantity significance of the beeswax and castor oil formulation.

Different types of wax were used to harden lipstick formulations. Patil et al. (2019), Bhagwat et al. (2017), Maru and Lahoti (2018), and Ghongade et al. (2021) used carnauba wax, while Lwin et al. (2020) use candelilla wax. Sunil et al. (2013) and Chaudhari et al. (2019) used paraffin wax [1,3,6,14,29,51]. All of these waxes are in categories as hardening agents, which are responsible for the hardness of the lipstick [108]. The differences between these waxes are their melting points: carnauba wax: 80 to 88 °C [17]; candelilla wax: ranging from 61 to 89 °C [21]; paraffin wax: various grades with different specified melting ranges are commercially available [108]. Carnauba wax is made from the leaves of the carnauba palm (*Copernicia prunifera*), which grows exclusively in the arid northeast of Brazil's Caatinga region (scrublands); it is a hard wax with the great melting point of natural wax and has a low solubility [109]. The natural carnauba leaves are coated with a waxy material which is the raw material for the production of carnauba wax [109]. Due to its high melting point, carnauba wax was suggested as a good base and moisturizer for the formulation of lipsticks [55,56]. Candelilla bushes (typically *Euphorbia cerifera, syn. Euphorbia antisyphilitica*) occur wild in northern Mexico and southern Texas, and candelilla wax is produced from their leaves and stems [110]. Boiling the plant material or extracting it with benzene are the two methods for obtaining candelilla wax, and it is commercially accessible in a yellow to brown hue [110]. It has been used in the food industry and in cosmetics and personal care products, especially lipsticks [55,56,110]. Alkenones wax was suggested as a promising base for lipstick and other personal care products [21]. It is an off-white waxy solid at room temperature and obtained from *Isochrysis* sp. [21].

Copra from the coconut palm (*Cocos nucifera*) is the dried kernel of the coconut, used to make coconut oil. It ranges in colour from white to a light brownish yellow. Coconut oil has a high concentration of low molecular weight saturated fatty acids, which are what distinguishes lauric oil from other oils [111,112]. The cholesterol-lowering, reduced risk of cardiovascular diseases (CVDs), weight loss, improved cognitive capabilities, antibacterial activity, and other health benefits of coconut oil have been reported [113]. It was concluded that the inclusion of coconut oil in lip preparations could soften, moisturize, and make lips look more healthy [114].

*Hylocereus polyrhizus* (pitaya) is a Cactaceae plant family that is known in Asia as "dragon fruit" [55]. It has the hue of the pulpy exocarp and/or the soft fleshy heart (mesocarp or endocarp), which carries the seed [55]. Pitaya has been linked to a variety of health benefits, including cancer prevention, anti-inflammatory and anti-diabetic properties, and cardiovascular mortality risk reduction [55]. As Pitaya contains a substantial amount of linoleic and linolenic acids, which are unsaturated fatty acids (UFAs), pitaya seed oil is normally included in natural lipstick formulations [55]. UFAs control the flow of oil through the skin and then improve the metabolism process within the skin [55].

Mangosteen rind (*Garcinia mangostana* L.) is considered to be a waste product and contains different types of water-soluble antioxidants. Alpha mangostin and other xanthones, which are found in the rind of mangosteen, have been shown to exhibit considerable antioxidant action [115]. The highest antioxidant activity was concluded for lipstick formulations prepared with 8 g of mangosteen rind extract [25].

## 4. Method of Preparation

Lipstick production varies very slightly depending on the type of ingredients used. The moulding method may be suggested as a standard procedure for the preparation of lipsticks (Figure 1). McIntosh et al. (2018) categorized the ingredients used in lipstick preparation as phase A, phase B, and phase C [21]. Waxes were represented in phase A, dyestuff and other oils were represented in phase B, and preservatives and other additives were represented in phase C [21]. The author heated up phase A to 80 °C, then phase B ingredients were added one by one to phase A. The mixture of phase A and B was removed from heat, and phase C was added and then poured in the lipstick moulds [21]. On a different occasion, the same method as above was followed by Esposito et al. (2021) to prepare organogel-based lipstick [116]. The only difference was that Esposito et al. (2021) heated phase A to 200 °C, phase B was heated separately to 100 °C, and the pigment was dispersed well in phase B and then added to phase A [116]. Typically, the wax phase (phase A) is created by melting the waxes in a water bath in decreasing order of the melting point. A suitable temperature should be used depending on the melting points of those waxes. The highest melting point of any wax constituent should be used to determine the water bath temperature used in the lipstick preparation [1,6,13,14,29,55].

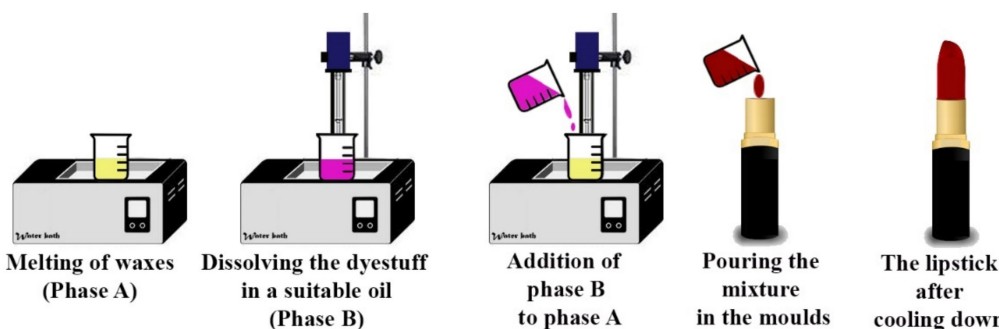

**Figure 1.** The general method of preparation for lipsticks.

## 5. Characterization

Maintaining a standardised criterion for herbal lipstick is essential. Hence, the formulated lipsticks were evaluated on parameters such as melting point, breaking point, thixotropy character, force of application, surface anomalies, aging stability, solubility test, pH parameter, skin irritation test, perfume stability, and lead limit test [14,29,48,50,51]. Furthermore, the Food and Drug Administration (FDA) controlled the manufacturing of lipsticks based on the detection limits of lead and colour additives [117].

### 5.1. Melting Point

Melting point determination is crucial for indicating the limits of safe storage [6,14]. Sunil et al. (2013) used the capillary tube method, using a digital melting point apparatus to determine the melting point of the formulated lipstick [14]. The method involved capillary filling and observed the temperature at which the lipsticks completely melted [14]. A similar method was used in several other studies [6,38,49,50,86,105].

The melting point test was described by Azwanida et al. (2014) and Bhagwat et al. (2017) in detail [6,49]. Both studies melted the ingredients and poured them into 50 mg glass capillary tubes opened at both ends [49] and cooled with ice for 2 h, and melting point was measured using a thermometer [6]. However, Azwanida et al. (2014) used a digital thermometer immersed in a beaker containing water placed on a hot plate stirrer [49]. Heating and stirring was started slowly at a fixed speed [49]. The measurement of melting point was observed when the lipstick sample moved along the capillary tube [6,49]. A digital melting point apparatus was also suggested for measuring the melting point of the lipstick [107]. Briefly, 2 g of lipstick sample was placed into a glass tube, and the tube was

dipped into a plate full of water, then heated in the water bath, and the melting point was noted [107]. The melting point of lipstick ranged from 59 to 64 °C [6,14,38,107].

### 5.2. Softening Point

The purpose of this test is to determine the ability of lipstick to withstand the range of conditions to which it will be subjected in the consumer's handbag [6,118]. Most of the studies reviewed in this paper have not conducted the softening point test. Bhagwat et al. (2017) used the ring and ball method by inserting the lipstick into a ring and placed it in the refrigerator (6 °C) for 10 min [6]. The ring and ball assembly was set and immersed in a water bath [119]. The temperature should rise at the rate of 1 °C per minute when reaching 45 °C [118]. The softening point was observed at the temperature at which the ball went through the lipstick sample [119]. The obtained softening point reported by Bhagwat et al. (2017) was 68 °C [6]. The target softening point ranged from 68 to 74 °C [119]. A higher softening point indicated better lipstick stability [119].

### 5.3. Breaking Point

Lipstick hardness is an important determination of the physical characteristic properties of a lipstick [48]. Jain and Sumeet Dwivedi (2017) used the breaking point test to determine the strength of a lipstick [48]. It was carried out by placing the lipstick horizontally in a socket $\frac{1}{2}$ inch away from a support edge [48]. At a 30 s interval, the weight of a fixed value of 10 g was progressively increased [48]. The breaking point was observed when the lipstick broke [48]. This method has also been used in several other studies [29,38,50,86,105]. The findings from this review indicated that no significant difference was obtained in the breaking point results of the studies. The breaking points ranged from 30 to 32 [29,38,50,86,105]. A texture analyser was also used to determine the breaking point of cocoa wax lipstick. The distance travelled by the hemispherical edge blade when a trigger force of 10 g was applied was used to determine the measurement [120]. The breaking of a lipstick was the end point of a test, and the values obtained in the texture analyser were recorded [120]. The study indicated that the higher the breaking point test numbers, the better texture [120].

### 5.4. Thixotropy Character

The thixotropy test is used to determine the uniformity in the viscosity of a base [48]. It is important to identify lipstick with good texture, which may ease application by consumers [121]. Jain and Sumeet Dwivedi (2017) used a penetrometer to determine the thixotropic properties of lipstick [48]. The authors briefly described the process by penetrating a needle under a 50-g load at 25 °C for 5 s [48]. The thixotropic character was observed by the depth of penetration [48]. This method was also used in several studies [14,38,50,53]. The thixotropy of lipstick ranged from 9 to 10.5 [14,38,48,50,53]. Based on the reviewed data, there is no single study that mentioned a comparison of the thixotropic results to the reference guidelines. This could suggest that no guidelines for this test have been developed.

### 5.5. Force of Application

A force of application test is used to evaluate the force required for lipstick application on the lips [50]. The method was described by Panda et al. (2018) [50]. The test was carried out by placing a piece of coarse brown paper on a shadow graph balance [50]. Then the lipstick was applied at a 45-degree angle to cover a 1 square inch area until fully covered [50]. The resulting force of application was shown by a pressure reading [50]. The same method was used by Sainath et al. (2016) [86]. This test was described in the reviewed data, and the results were based on the authors' experience and comparison with other published data. To our knowledge, there are no specific guidelines on how this test is to be followed or how it is to be used as a standard reference.

*5.6. Surface Anomalies*

A surface anomalies inspection is important for product quality control and to meet the customers' expectations [122]. The visual inspection method was used to assess all anomalies which can occur on the surface of lipstick [14,29,51,86,123]. The lipstick was evaluated with the following standards: [124]

"Mark": something that damages the surface, a break in the form (for example, scratches, dent, etc.) [124].
"Heterogeneity": anything that makes the product lose homogeneity (for example, stain, difference in colour and/or texture, etc.) [124].
"Pollution": anything that appears added to surface and is considered undesirable (for example, dust, particles, etc.) [124].
"Distortion": anything that changes the shape of the surface (for example, an irregular line, etc.) [124]. Freedom from any defects such as no formation crystals and/or no contamination by moulds and fungi on lipstick surfaces indicate that the lipstick has achieved the required quality [12,29,51,123].

*5.7. Stability upon Storage*

An assessment of stability upon storage needs to be conducted to determine product storage requirements as well as to establish acceptable parameters for products and package expiration dates [29]. The visual inspection method was used to evaluate organoleptic characteristics including colour, odour, pH, and appearance, such as bleeding (separation of coloured liquids from the waxy base that leads to uneven colour distribution) [125] and crystallization on the surface of lipstick [6,29]. The stability test of a lipstick begins 48 h after formulation [49]. Accelerated stability studies were carried out [6,49,126] by storing 350 g of lipstick samples at room temperature (24.0 ± 3.0 °C) for 48 h and then evaluating baseline (t0) properties [49]. Lipsticks were then stored under three different temperatures: refrigerator temperature (4 °C) [6], room temperature (24.0 ± 3.0 °C) [6,49], and high temperature (40.0 ± 2.0 °C) [6,49] for 1 h [6,29,49]. Lipsticks were assessed on the 3rd, 7th, 15th, 30th, and 60th days [6,49]. Susmiatun (2018) assessed for one month by observing on the 1st, 5th, 10th, 15th, 20th, 25th, and 30th days [126]. Assessments at t0 were considered as a reference with which to compare the results [49]. The accelerated stability studies were carried out as per the ICH guidelines by storing at 40 °C/75% relative humidity [127].

*5.8. Spreadability Test*

A spreadability test is used to determine the ability of lipstick consistency to spread on a surface [128]. This test was conducted in two previous studies in which a lipstick was applied for at least 3 cm onto a paper [129] or a glass slide [130]. Then the smoothness and uniformity of a protective layer formation from the lipstick was visually observed. Additionally, the lipstick was evaluated with the following standards: [129,130].

Excellent (E): No fragment, smooth and uniform surface application without deformation of lipstick [129,130].
Intermediate (I): Few fragments, uniform application with little deformation of lipstick [129,130].
Unsatisfactory (U): Many fragments, not uniform application with intense deformation of lipstick [129,130].

*5.9. Solubility Test*

Solubility tests can suggest the polarity of the compound in lipsticks in order to characterize the solvent selectivity [131–133]. The method described by Maru and Lahoti (2018) involved adding a few drops of lipstick sample to different solvents—methanol, ethanol, chloroform, and petroleum ether—in different test tubes, and the solubility was observed [51]. This method was also used in several studies [14,38,50,86,105]. The lipstick was soluble in chloroform and ethanol [14,38,50,86,105]. Lipstick containing castor oil is

soluble in alcohol [18] and has limited solubility in petroleum solvents due to the hydroxyl group in ricinoleic acid [134]. Nonetheless, no research article has focused on the solubility significance of a lipstick formulation.

### 5.10. pH Parameter

The pH stability profile and the safe pH range for lipsticks define the acceptable limits of the products to be safely applied on the lips [135]. The pH level affects the solubility of ingredients, which can alter the physical and microbiological stability of a product [136]. Extreme pH can damage the skin barrier [136]. The pH of healthy lips is on average 4.7 [137]. A potentiometric method using a pH meter apparatus was carried out to determine the pH of formulated lipsticks [14,86,138,139]. One author briefly described the process by melting the lipstick in a water bath and noting the pH [140]. However, the pH requirement for lipstick that was safe to apply on lips ranged from pH 4 to 7 [14,86,139,140]. This combination of findings from the reviewed literature provides no significant difference in the obtained pH results which ranged from 4 to 6.5 [6,14,38,86,107].

### 5.11. Skin Irritation Test

Human models were utilised in certain studies on skin irritation tests for lipstick. For instance, a study carried out by Panda et al. in 2018 used humans as animal models. The prepared lipstick was applied to the skin (lip). Any symptoms such as itching, irritation, and redness were observed for a duration of 10-min [50,86]. A mouse model was also suggested to be used for the skin irritation test of the lipsticks. Mice were separated into two groups and subjected through a test protocol, with each group consisting of five mice for each test sample. Each mouse was mildly anaesthetized with chloroform and had approximately 10 mg of lipstick samples placed to the dorsal region of their left ears for the test groups [1]. For any skin irritation on the treated area, the test group of mice was compared with the control group [1]. It is worth mentioning that cosmetic testing on animals is not recommended [141] and that it is banned in some countries [142]; alternative appropriate and effective methods to examine skin irritation are already available to replace animal testing [143].

### 5.12. Lead and Other Metals Limit Test

Several studies conducted research on the determination of the total lead in the lipstick and other cosmetic preparations [144]. Moreover, the FDA undertook a project to evaluate the amount of lead in lipstick and a range of other cosmetics as part of its overarching goal to protect public health [117]. The FDA guidelines stated, "FDA scientists developed and validated a new method for analysing lead in lipsticks and used this method to find the lead content in several hundred cosmetic lip products on the U.S. market, including 20 from CSC's report that were still available. We used a more common extraction method to find the lead content in additional cosmetic lip products. We used both of these methods to find the lead content in a total of 685 cosmetics on the U.S. market" [117]. Using the flame atomic absorption spectrophotometry method, the levels of lead, nickel, copper, zinc, and iron were analysed and detected [9]. Determination of lead in the lipstick and hair dyes was assessed using a novel microwave-assisted dispersive liquid–liquid microextraction (MADLLME) technique and graphite furnace atomic absorption spectrometry (GFAAS) [145]. As a result, the newly discovered technology was effectively used to extract and analyse lead ions in lipsticks and hair dyes [145].

## 6. Conclusions

Studies reported in this review indicated that despite the high demand for lipsticks in the market, consumers need to take precautions, as regular use of synthetic-based lipsticks might have high potential to cause health problems, including skin irritation and allergy reactions. These findings support the need to better understand the role of synthetic and natural ingredients in the production of the lipsticks. Existing evidence noted that the use

of a mixture of waxes with varying melting points and controlling the ultimate melting point of the stick by adding an appropriate amount of a high melting point wax yields the best results. According to the literature, there was a little discrepancy in the method of lipstick preparation, indicating that the "moulding method" should be used as a generic method. The findings of this review have a number of important implications for future practice, including a call for extensive testing and clinical trials in order to assess the efficacy and established safety profile in the formulation of lipsticks. A key policy priority should therefore be in place to plan for the development of specific guidelines for the formulation and characterization of lipsticks.

**Author Contributions:** Conceptualization, S.M.M.; methodology, S.M.M. and N.A.B.A.R.; software, S.M.M., N.A.B.A.R. and N.O.; validation, N.O., S.M.M. and S.O.A.; formal analysis, N.A.B.A.R.; investigation, S.M.M. and N.A.B.A.R.; resources, N.O., S.M.M. and S.O.A.; data curation, S.M.M., N.A.B.A.R. and N.O.; writing—original draft preparation, N.A.B.A.R.; writing—review and editing, N.O., S.M.M. and S.O.A.; visualization, S.M.M. and N.O.; supervision, S.M.M. and N.O.; project administration, S.M.M., N.A.B.A.R. and N.O. All authors have read and agreed to the published version of the manuscript.

**Funding:** This research received no external funding.

**Conflicts of Interest:** The funders had no role in the design of the study; in the collection, analyses, or interpretation of data; in the writing of the manuscript; or in the decision to publish the results.

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
