# Peer review of "Lipsticks History, Formulations, and Production: A Narrative Review"

_cosmetics, doi:10.3390/cosmetics9010025_

Round 1
Reviewer 1 Report
Comments are attached

Author Response
Dear reviewer, we would like to show our gratitude for your time that you spent and efforts to improve the quality of this paper. Please find the amendments that we have done based on your valuable comments in the following table:
|
Reviewer 1 |
|
|
1. |
“Medicinal value” paid attention in the Introduction, but correlation with reviewed data later is missing. |
|
|
|
|
|
We have added the following sentences:
i. Page 5, Line 139 Therefore, Vitamin E has been used in lipsticks preparations as an antioxidant
ii. Page 5. Line 141
iii. Page 6. Line 211
|
|
2. |
“History of lipstick” should be improved and rewritten in scientific level. |
|
|
|
|
|
The “history of lipstick” was rewritten in scientific level and the information was arranged chronologically (Page 2, Line 51 to 65) |
|
|
|
|
3. |
“Ingredients of lipstick” should be expanded: separate ingredients to natural and synthetic (1 table); add information about herbal active substances used in formulations. All part is focused on laboratory scale, but industrial formulations are also. |
|
|
Thanks for highlighting these points. |
|
|
|
|
|
|
|
4. |
“Method of preparation” contains one method, but differences or similarities of cited researchers is not clear. I recommend to present this part in graphical format. |
|
|
|
|
|
This part was rewritten, and graphical format was added. (Page 8 and 9. Line 358 to 375) |
|
5. |
“Characterization” part should be supplemented: what instrument were used in reviewed studies; what official requirements can be set as quality parameters for lipsticks. |
|
|
|
|
|
The characterizations were amended and information on “Skin irritation test” and “Lead and other metal limit test” was added, (Page 11, Line 500 to 525)
|
|
|
|
|
6. |
“what instrument were used in reviewed studies?” |
|
|
|
|
|
We added the following information: i. Page 9, Line 386
ii. Page 9, Line 403 to 405 “…ring and ball method……in a water bath”
iii. Page 9, Line 418 “The texture analyser was also used….”
iv. Page 10, Line 426 “….used penetrometer to determine the thixotrophy properties of the lipstick”
v. Page 11, Line 493 “…using pH meter…”
|
|
|
|
|
7. |
What official requirements can be set as quality parameters for lipsticks? |
|
|
|
|
|
We added the following sentences.
i. Page 10, Line 430 Based on the reviewed data, there is no single study mentioned the comparison of the thixo-trophic results to the reference guidelines. This could suggest that no guidelines for this test have been developed.
ii. Page 10, Line 438 This test was described in the reviewed data and the results were based on the authors experience and comparison with other published data. To our knowledge, there is no specific guidelines for this test to be followed or used as standard reference. iii. Page 10, Line 467 The accelerated stability studies were carried out as per the ICH guidelines by storing at 40 ºC/75% relative humidity
|
|
|
|
|
8. |
Please improve the conclusion chapter. |
|
|
|
|
|
We have edited the conclusion as suggested by the reviewer (Page 11, Line 528) |
|
|
|

Reviewer 2 Report
The article is well structured and didactic, the subject is very interesting, but the information is presented in a very superficial way.
I understand that it may be a little explored topic in the scientific literature, but the type of article - review, has a lower content than other examples in the same journal.
I recommend that the topic be described in more depth, for example on the pharmacological properties that can be incorporated into lipstick.
Author Response
Dear reviewer, we would like to show our gratitude for the time that you spent and your efforts to improve the quality of this paper. Please find the amendments that we have done based on your valuable comments in the following table:
|
Reviewer 2 |
|
|
|
The article is well structured and didactic, the subject is very interesting, but the information is presented in a very superficial way. I understand that it may be a little explored topic in the scientific literature, but the type of article - review, has a lower content than other examples in the same journal. |
|
|
|
Thanks for the comments and suggested improvements. The article was rewritten, and more detailed information was provided |
|
|
|
I recommend that the topic be described in more depth, for example on the pharmacological properties that can be incorporated into lipstick. |
|
i. Page 4, Line 91 ii. Page 4, Line 116 iii. Page 4, Line 132 iv. Page 5, Line 150 v. Page 6, Line 208 vi. Page 6, Line 232 vii. Page 6, Line 238
viii. Page 6, Line 252 ix. Page 7, Line 262 “Curcuma longa (turmeric) was utilized….”
x. Page 7, Line 287 xi. Page 8, Line 340 “Cholesterol lowering, reduced risk of cardiovascular….”
xii. Page 8, Line 345 “It has the hue of the pulpy exocarp…”
xiii. Page 8, Line 347
|
|
|

Reviewer 3 Report
Line 15; should be written either To date or to our knowledge.
Line 50-51; should be rewritten.
Table 1, please separate references by a comma.
Line82; please write the full name of the castor oil.
Author Response
Dear reviewer, we would like to show our gratitude for the time that you spent and your efforts to improve the quality of this paper. Please find the amendments that we have done based on your valuable comments in the following table:
|
|
Reviewer 3 |
|
|
|
|
1. |
Line 15; should be written either to date or to our knowledge. |
|
|
|
|
|
Line 15 was changed “to date” and “to our knowledge was removed”. |
|
|
|
|
2. |
Line 50-51; should be rewritten. |
|
|
|
|
|
Line 50-51 was rewritten, please check the line number (47-49) which highlighted in yellow colour in the new attached manuscript. |
|
|
|
|
3. |
Table 1, please separate references by a comma. |
|
|
|
|
|
The references were separated by a comma |
|
|
|
|
4. |
Line 82; please write the full name of the castor oil. |
|
|
|
|
|
The full name of the castor oil was added (Line 301) |

Round 2
Reviewer 1 Report
The manuscript is improved and suitable for publication.
Author Response
Thank you very much.
Reviewer 2 Report
Complementary information was added to the text, but it was extensive, tiring and non-didactic.
Animal testing is no longer accepted for cosmetics. The paragraph starting at line 498 is completely at odds with ethical principles. Alternative methods already exist to assess skin irritation (OECD 439) or biocompatibility (ISO10993).
Author Response
Thanks for highlighting this point. We have added the following sentences. ( Page 11, line 507-509)